# Building-Level Urban Functional Area Identification Based on Multi-Attribute Aggregated Data from Cell Phones—A Method Combining Multidimensional Time Series with a SOM Neural Network

Zhenglin Song [1] , Hong Wang [1,*], Shuhong Qin [2], Xiuneng Li [2], Yi Yang [1], Yicong Wang [1] and Pengyu Meng [1]

1    School of Hydrology and Water Resources, Hohai University, Nanjing 210098, China;
     zlin_song@hhu.edu.cn (Z.S.); yiy125@hhu.edu.cn (Y.Y.); wangyicong@hhu.edu.cn (Y.W.);
     mengpengyu@hhu.edu.cn (P.M.)
2    School of Earth Science and Engineering, Hohai University, Nanjing 210098, China;
     qinshuhong@hhu.edu.cn (S.Q.); xiuneng@hhu.edu.cn (X.L.)
*    Correspondence: hongwang@hhu.edu.cn

**Abstract:** Portraying functional urban areas provides useful insights for understanding complex urban systems and formulating rational urban plans. Mobile phone user trajectory data are often used to infer the individual activity patterns of people and for functional area identification, but they are difficult to obtain because of personal privacy issues and have the drawback of a sparse spatial and temporal distribution. Deep learning models have been widely utilized in functional area recognition but are limited by the difficulty of acquiring training samples with large data volumes. This paper aims to achieve a fast and automatic identification of large-scale urban functional areas without prior knowledge. This paper uses Nanjing city as a test area, and a self-organizing map (SOM) neural network model based on an improved dynamic time warping (Ndim-DTW) distance is used to automatically identify the function of each building using mobile phone aggregated data containing work and residence attributes. The results show that the recognition accuracy reaches 88.7%, which is 12.4% higher than that of the K-medoids method based on the DTW distance using a single attribute and 7.8% higher than that of the K-medoids method based on the Ndim-DTW distance with multiple attributes, confirming the effectiveness of the multi-attribute mobile phone aggregated data and the SOM model based on the Ndim-DTW distance. Furthermore, at the traffic analysis zone (TAZ) level, this paper detects that Nanjing has seven functional area hotspots with a high degree of mixing. The results can provide a data basis for urban studies on, for example, the urban spatial structure, the separation of occupations and residences, and environmental suitability evaluation.

**Keywords:** urban functional areas; mobile aggregated data; dynamic time warping; SOM

## 1. Introduction

Urban functional areas were first proposed in the Athens Charter, which called for planners to deal with four types of urban areas: residential, work, recreational, and transportation areas [1]. As the city continues to grow, the emergence of other functional areas makes the spatial structure of the city more complicated [2,3]. The type of these functional areas can be defined by the activities or spatial interactions that may occur in the area [4,5]. The study of urban functional areas is conducive to the rational and sustainable planning of future cities and the efficient and adequate use of urban space [6,7]. At the same time, the discovery of urban functional zones facilitates various aspects of human life, health and transportation [8–10]. Therefore, it is crucial to effectively collect information on different functional areas of cities. Methods based on conventional remote sensing (RS) techniques and high spatial resolution (HSR) remote sensing imagery have been widely used to extract and analyze urban land use and land cover because of their ability to capture the natural

appearance of the land surface [11–14]. However, urban functional areas are usually more concerned with the socioeconomic attributes within the area, and such attributes are difficult to distinguish from remotely sensed images, which are good at responding to the physical properties of the surface of features (e.g., reflectance and texture) [15].

With the development of mobile location services and cloud processing technologies, social sensing big data that record human daily life, such as point of interest (POI) data [16,17], social media data [18,19], taxi trajectory data [20], street view data [21] and cellphone location data [11,22], which contain rich socioeconomic attributes, are widely used in the delineation of urban functional areas. For example, Yao et al. (2017) used POI data with rich semantic and location information to extract the high-dimensional feature vectors of POIs within a TAZ to identify urban land use types [16]. The information contained in POI data can be used to describe the quantity and distribution of each land use type in the area, but it is not possible to use the information contained in these data to directly classify the functional areas of a city. For example, restaurants located in residential areas mainly serve the daily needs of nearby residents, while large restaurants located in commercial areas play the role of attracting visitors from all over the world. POIs with similar functions may be classified into different functional areas due to their different locations. There is a strong correlation between the function of an area and the behavioral activities of the people visiting it [23,24]. If we combine rich semantic POI data with new geographic data reflecting residents' behavior and activities, such as residents' individual activity trajectory data and location-based social network user check-in information, we can more accurately identify urban functional areas [5,18,25]. Compared with data from social media, taxi trajectory and public transportation card, cell phone data have the advantages of a large number of users, a wide spatial coverage and strong passive access, which can reflect the activities and behavior of the urban population in a more objective and comprehensive way.

Mobile phone location data include two types of user trajectories based on locations and aggregated data based on base stations. Trajectory data are widely used to infer individual activity patterns [26,27], estimate the population distribution at fine spatial and temporal scales [22,28] and classify urban functional areas in combination with social media data [19] due to their complete population trajectory information. Since trajectory data involve personal privacy issues, it is difficult to obtain such data, and these data are recorded only when events such as calling, texting or moving location occur, which has the defect of a sparse spatial and temporal distribution [29]. In contrast, cell phone aggregation data are automatically and quickly recorded when residents move to the coverage area of mobile communication base stations and make requests for location-based services. Due to the high spatial distribution density of base stations within cities, the spatiotemporal behavior of residents' activities can be well captured [30]. Aggregated data do not involve personal privacy or security; thus, they often provide collateral information about mobile users, such as their gender, age and work attributes, which can improve the refinement of the description of crowd activity patterns and help to better understand the urban spatial structure. For example, Zhan et al. (2013) described the spatial characteristics of residential conditions and employment of Beijing residents based on social attributes such as the age, education, and income of urban residents [31]. Xiao et al. (2017) further explored the correlation between the distribution of parks and the spatial clustering of different social groups (e.g., different ages, different income levels) [32].

How to use the socioeconomic activity behaviors of the population to infer the functions of urban areas is the key to functional area classification. Probabilistic topic models (PTMs) are often used to infer urban functions [18,33] but suffer from the drawbacks that the algorithms are more time consuming and are very sensitive to prior knowledge and the fine-tuning of parameters. The DTW distance-based k-medoids model is the main method for the functional zoning of cities using social media data and POIs because it is directly driven by raw data and has a low computational cost [34]. The DTW method can only calculate the distance between the time series data of one-dimensional attributes and can be improved to solve the problem of multidimensional attribute data [35]. On the other hand,

the results of clustering algorithms based on k-means or k-medoids are often influenced by the *k*-value and initial cluster centers. The model can only acquire the surface meaning of the data during the clustering process and cannot learn the implicit high-dimensional features [36]. Therefore, the LSTM network approach is used to mine crowd activity patterns for urban functional area identification [37]. However, LSTM network models require a large amount of labeled data as training samples, which limits the large and fast functional area identification in urban areas. Deep learning-based clustering methods, such as self-organizing map (SOM), do not require training samples and obtain a low-dimensional representation of each data point for network learning and clustering through network self-optimization [38]. SOM has been widely used in pattern clustering, speech recognition and other research [39,40]. Wandeto et al. (2018) used the SOM method to detect potential changes in environmental conditions or the urban landscape structure in a large time series of image data and showed that the method can quickly and effectively identify changes in the Las Vegas urban landscape during the 1984–2008 period [39]. However, no research has been reported on the use of the SOM method to distinguish population activity patterns and to then delineate urban functional areas. Therefore, this paper aims to investigate whether the SOM method based on the multidimensional DTW distance can effectively identify urban functional areas without training samples. Specifically, this paper takes Nanjing city as the experimental area, takes each building as the research object and uses aggregated cell phone data with both work and residence attributes and the SOM method based on the Ndim-DTW distance to infer the functional type of each building. The data are further aggregated to the TAZ level to detect potential urban functional area hotspots. The accuracy of cell phone data with or without attribute information and the classification results of different clustering methods are compared, and the accuracy of the detected functional area hotspots is qualitatively assessed. Finally, the relationship of functional area hotspots with urban planning and population density is separately discussed.

## 2. Study Area and Datasets

### 2.1. Study Area

Nanjing is located in the eastern part of China, in the middle of the lower reaches of the Yangtze River, with a total area of 6587 square kilometers and a built-up area of 971.62 square kilometers (http://tj.jiangsu.gov.cn/2020/nj18/nj1805.htm, accessed on 18 November 2021). As an important central hub city in eastern China, Nanjing has a high level of economic and cultural development, a complex urban pattern and highly mixed land use types. In this paper, nine administrative districts in Nanjing are selected as the study area, including the six main urban areas of the Gulou, Xuanwu, Jianye, Qinhuai, Qixia, and Yuhuatai districts, as well as the Jiangning, Lishui, and Pukou districts (Figure 1).

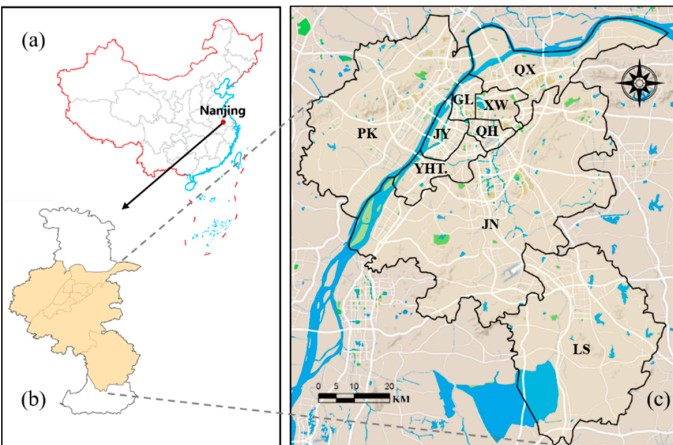

**Figure 1.** Location of the study area. (**a**) Nanjing, China. (**b**) The study area. (**c**) The extent of eight districts with Google Maps as the background.

### 2.2. Data

We used the following five urban datasets to identify urban functional areas:

- Mobile Subscriber Data. The mobile subscriber data covering nine districts in Nanjing (Figure 1c) were purchased from the Jiangsu Mobile Company and acquired from 18–24 February 2019. with an hourly temporal resolution and a 150 m × 150 m grid spatial resolution. Whenever a cell phone makes a communication connection with a base station (such as receiving calls, sending and receiving SMSs, location updates), the base station automatically records and generates signaling data containing the base station location information. The purchased mobile subscriber data come with users' work attributes and residence attributes. The user attributes are judged based on the range of time periods and the length of time that the user stays in the base station coverage area. The specific discrimination method is as follows: when a user appears in the base station coverage area for more than 7 days in a month and the time of appearance is from 10:00 to 17:00, the user is assigned the work attribute; when a user who appears in the base station coverage area for more than 7 days and the time of appearance is from 0:00 to 6:00 or 21:00 to 23:00, the user is assigned the residence attribute. The number of cell phone users, residential attribute users and work attribute users in each base station area in each hour is statistically obtained, and the population data of these three attributes are interpolated into a 150 m × 150 m grid. Each grid includes the grid ID, time (where 2019 represents the year, 0218 represents 18 February, and 0100 represents 1:00 a.m.), latitude and longitude, and the number of people with different attributes (Table 1).

**Table 1.** The format of mobile user data.

| ID | Region | Time | Longitude (°) | Latitude (°) | No. of People | | |
|----|--------|------|---------------|--------------|-----|-------------|------|
| | | | | | All | Residential | Work |
| 1 | Q.H. | 201902180100 | 118.75767 | 32.062205 | 1372 | 768 | 104 |

- Building data. The vector data of buildings in the main urban area of Nanjing were mainly obtained by downloading from the BIGEMAP platform (http://www.bigemap.com/, accessed on 18 November 2021). The missing building data around suburbs were obtained by overlaying with GF-2-urban images and were then visually interpreted. The GF-2-urban images were obtained after intercepting the GF-2 image with the impervious surface distribution in Nanjing (from the website http://data.ess.tsinghua.edu.cn/, accessed on 18 November 2021) as the built-up area [41]. Impervious surface data refer to surfaces such as roofs, asphalt pavements or concrete pavements, and in this study, we use the 2018 impervious surfaces extracted from Landsat images using the "exclude and include" framework [42] to represent the extent of the built-up area of Nanjing. GF-2 data were downloaded from the Land Observing Satellite Data Service platform (http://36.112.130.153:7777/DSSPlatform/index.html, accessed on 18 November 2021) on 23 May 2019. GF-2 data contain panchromatic (PAN) images with a resolution of 0.89 m and multispectral (MSS) images with a resolution of 3.2 m. The MSS data were subjected to RPC orthorectification [43], radiometric calibration and FLAASH atmospheric correction [44], and the data were fused with the RPC-orthorectified PAN data using the nearest neighbor diffusion method [45]. Finally, there were 122,544 building polygons in the built-up area of Nanjing (Figure 2), which were used as the basic analysis units for the subsequent clustering of urban functional areas;

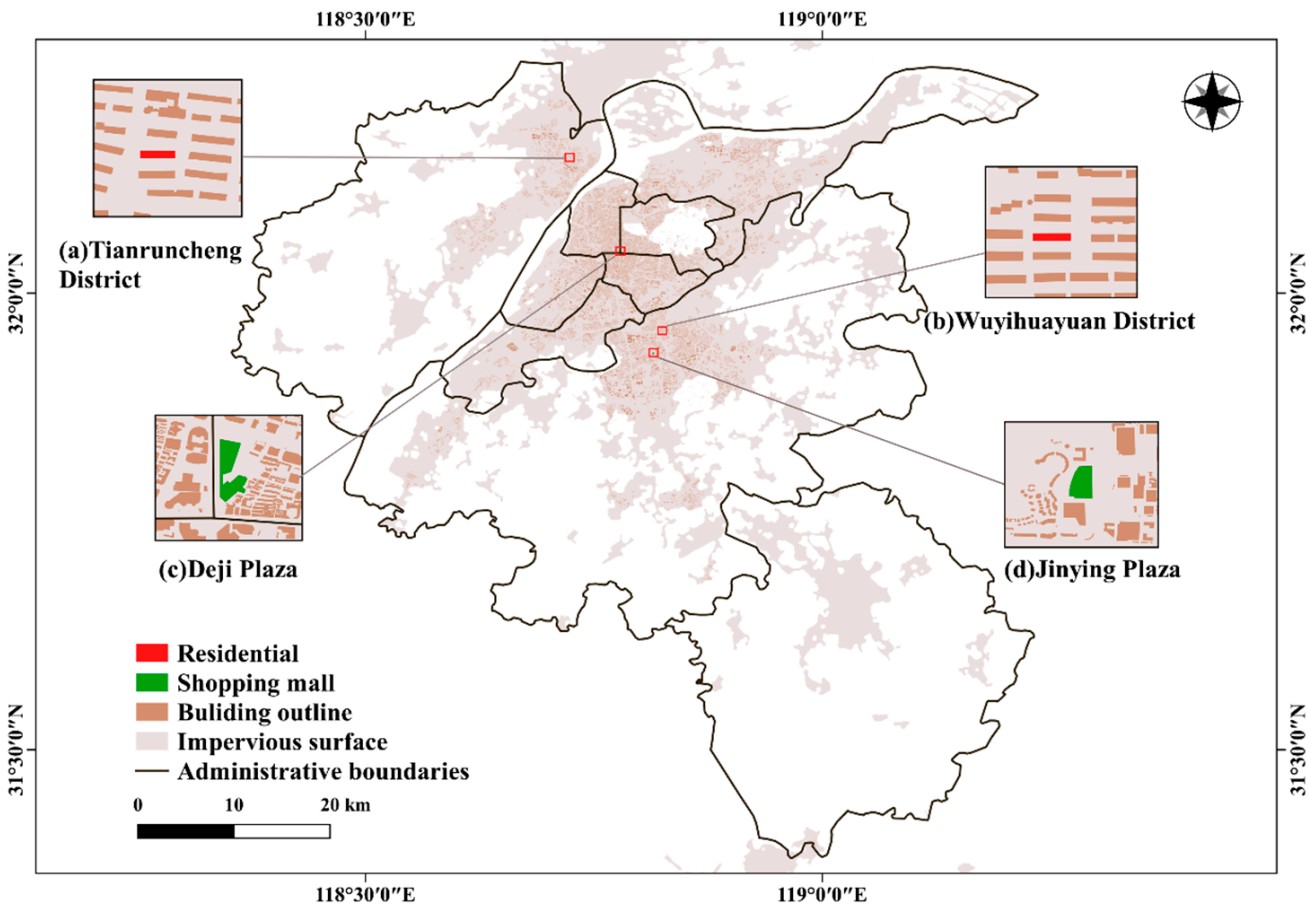

**Figure 2.** Building outline data in Nanjing city: (**a**,**b**) are residential buildings in two districts; (**c**,**d**) are two shopping malls.

- POI data. The Gaode Map Service POI data covering the study area were acquired in December 2018 and purchased from Gaode (https://lbs.amap.com, accessed on 17 January 2022), one of the largest web mapping service providers in China. POI can represent all places with location, which have large or small spatial scope and high or low recognition, but not all POI can provide effective information for building function speculation, or even cause interference, so points with small spatial granularity and low public recognition, such as public toilets, bus stops, newsstands, etc., need to be eliminated from the original data first [46]. Then, the remaining POI points were reclassified according to the building function type, referring to Gong et al. (2019) for the basic urban land use classification criteria in China [47,48]. The acquired POI data were regrouped into nine categories, namely: residential; business; shopping malls; industrial; administrative; medical; parks and greenspace; educational; and public facilities (Table 2). Considering the specificity of the data and study area, our POIs categories are slightly different from the classification system; for example, the commercial category was divided into shopping malls and business because our data can finely capture the activities of people with both work and residential attributes, and education was separately distinguished from public facilities because Nanjing is rich in educational resources. Since the number of different POI categories varies greatly and the spatial distribution of the same land use types is uneven, the original POI data need to be reconstructed to eliminate the bias of the data. The reconstruction methods mainly include the following: (1) In response to the problem of commercial POIs being repeatedly marked, we removed commercial POIs with a distance of less than 10 m; (2) since the number of industrial and residential POIs had been underestimated, industrial and residential POIs were added according to the method proposed by

Zhang et al. [49]; and (3) for the public category, POIs were added on the buildings by visual interpretation due to the relatively lower classification accuracy via the above method. The amount of reconstructed POI-constructed data increased significantly (Table 2), and various land use types were more evenly distributed in space. The POI data reconstruction method and the spatial distribution of the reconstructed POI data are referenced [41]. The POI-constructed data are used to annotate building functional types.

**Table 2.** POI categories and comparison of the number of original POIs and reconstructed POIs.

| Primary Categories | POI Labels | Original | Regenerated |
|---|---|---|---|
| Residential | Residential, Villa, Real estate subsidiary | 60,341 | 82,770 |
| Business | Business building, The investment company, Bank, Securities company, Financial company, Insurance company | 25,802 | 25,802 |
| Shopping malls | Shopping area, Food, Entertainment, Hotel | 91,308 | 81,702 |
| Industrial | Factory, Industrial | 2594 | 13,961 |
| Administrative | Foreign institutions, Government agencies, Public security organs, Scientific research institutions, Social groups, The tax agency | 10,524 | 17,142 |
| Educational | University, Educational school affiliation, Kindergarten, Middle school, Primary school, Vocational and technical school | 9591 | 14,805 |
| Medical | Clinic, General hospital, Health care subsidiary, Pet hospital, Specialized hospital, Plastic surgery hospital, Psychiatric hospital | 8049 | 11,498 |
| Sports and cultural | Museum, Archives, Convention and exhibition center, Science and technology museum, Gallery, Cultural center, Exhibition hall | 5827 | 8434 |
| Parks and greenspace | Park, Mountain | 2554 | 7685 |

- Traffic Analysis Zone. The traffic analysis zone is generated by overlaying the OSM road network buffers with the impervious surface data of Nanjing city. In this study, the OSM road network data of Nanjing city for December 2018 were downloaded from OpenStreetMap (https://www.openstreetmap.org, accessed on 18 November 2021). OSM was divided into seven classes and set different buffer radii, which were obtained by counting the actual road radius. For example, primary was set to 44 m, secondary was set to 34.8 m, tertiary was set to 30.4 m, residential was set to 21.5 m, motorways were set to 42 m, trunks were set to 60.5 m and railways were set to 7.7 m, as detailed in [41]. After overlaying, the final 8209 traffic analysis zones (TAZ) were obtained;
- FROM-GLC10. FROM-GLC10 is the world's first 10 m resolution global land cover map [50], and it can be downloaded for free from this website (http://data.ess.tsinghua.edu.cn/, accessed on 18 November 2021). In this study, FROM-GLC10 data were intercepted with the boundaries of the study area to obtain land cover types in Nanjing, including water bodies, grasslands, drylands, and woodlands.

## 3. Method

The research method consists of the following four steps (Figure 3): In the first step, the mobile subscriber data are preprocessed, including the estimation of the kernel density to obtain the attribute population density and the generation of attribute population two-dimensional time series curves; in the second step, the SOM network model is constructed, and the patterns are clustered; in the third step, the clustering results are overlaid with the reconstructed POI data to classify the individual building functions and to identify TAZ-level urban functional hotspots; fourth, the accuracy of the results is validated.

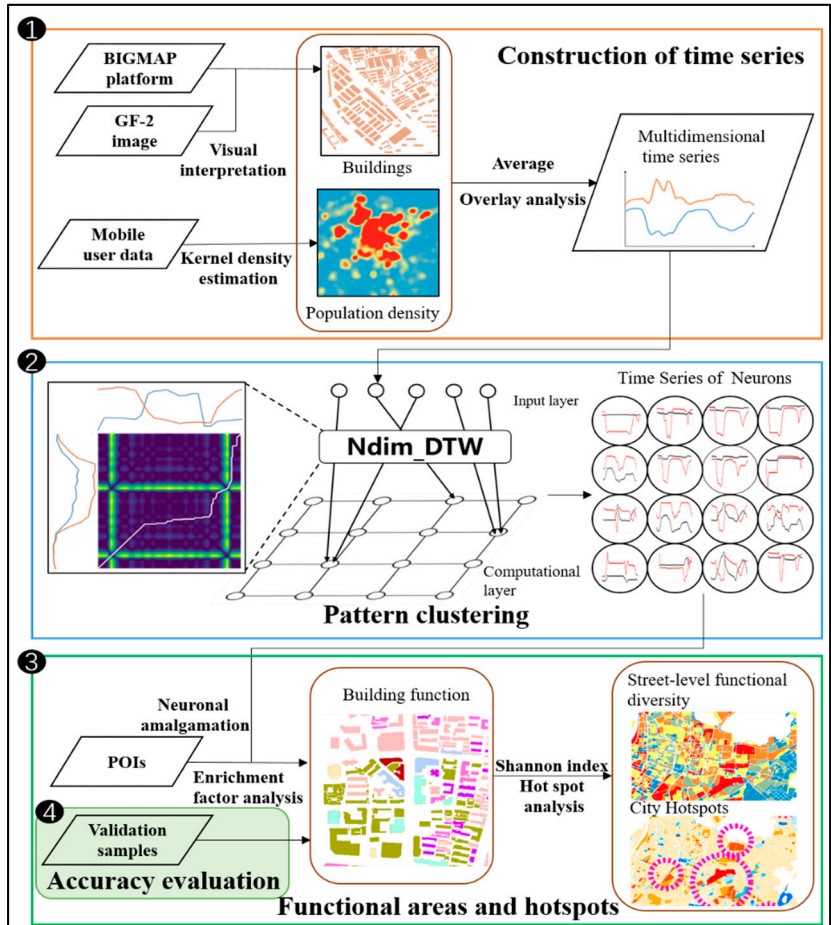

**Figure 3.** The workflow of identifying functional areas in Nanjing city.

### 3.1. Mobile User Density Dataset with Attributes

First, to ensure that crowd information is captured on each building, discrete grid data is converted to continuous user density raster data using Kernel Density Estimation (KDE), which is a nonparametric method for estimating probabilistic density functions to calculate the density of elements in their surrounding neighborhoods [51]. The equation for KDE can be expressed as:

$$f(x) = \frac{1}{nh^d} \sum_{i=1}^{n} K\left(\frac{1}{h}(x - x_i)\right) \tag{1}$$

where $K()$ is the kernel function; $h$ is the bandwidth or search radius; $n$ is the number of known points within the bandwidth; and $d$ is the dimensionality of the data.

We set the search radius to 2 km and these data are hereafter referred to as the mobile user density ($MUD$) dataset, which includes the mobile user density ($MUD_m$), residential user density ($MUD_r$) and working user density ($MUD_w$).

Then, the vector data of each building are overlaid with the $MUD$ dataset for analysis to obtain the dynamic change curve of the population density for each building. Figure 4 shows the $MUD$ change curves over time within two typical residential buildings (Figure 2a,b) and two shopping mall buildings (Figure 2c,d), and it can be seen that the buildings with the same function have similar $MUD$ change curves. In addition, there is a clear periodicity in the weekday $MUD$ variation curves for both residential and shopping mall buildings, but it is different from the weekend variation curves. To reduce the computational burden, the $MUD$ data are averaged by weekdays and weekends [52].

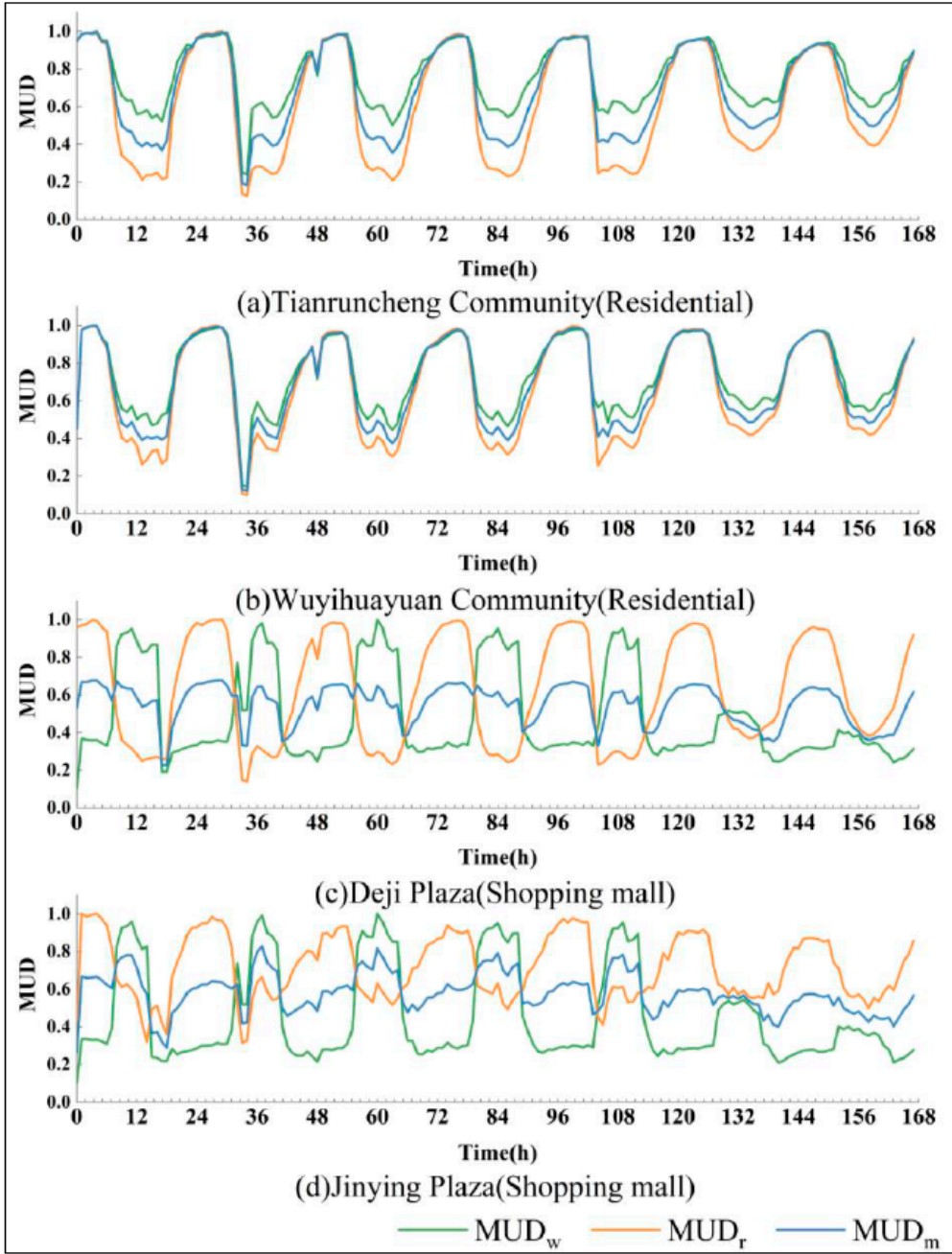

**Figure 4.** Time series of the total user density, residential population density and working population density for the four buildings in Figure 2a–d.

### 3.2. Ndim-DTW Distance Algorithm

The dynamic time warping (DTW) algorithm can measure the similarity of time series [53]. The DTW distance is the length of the best alignment (i.e., warping path) between two given time series, and the larger the DTW distance is, the more significant the difference between the two time series is. The traditional DTW distance is used to study one-dimensional sequence data. To describe the two attribute datasets of this study, residence and work, the DTW algorithm is improved, and the Ndim-DTW algorithm is constructed to process the multidimensional data.

In this study, each building object in the dataset is a multidimensional time series. The following definitions of key terms are given.

**Definition 1.** *A time series $T = t_1, t_2, \ldots, t_n$ is an ordered set of real values. The total number of real values is equal to the length of the time series.*

**Definition 2.** *A multidimensional time series (MDT) consists of M individual time series ($M \geq 2$), where each time series has n observations.*

$$T_1 = t_{1,1}, t_{2,1}, \ldots, t_{n,1} \tag{2}$$

$$T_2 = t_{1,2}, t_{2,2}, \ldots, t_{n,2} \tag{3}$$

$$\ldots$$

$$T_M = t_{1,M}, t_{2,M}, \ldots, t_{n,M} \tag{4}$$

To better understand the method, the calculation of the Ndim-DTW distance is illustrated with an example involving two types of building objects ($b_P$ and $b_Q$). In this example, the *MUD* time series of $b_P$ and $b_Q$ can be represented as $MDT_P : \{P_1 = p_{1,1}, p_{2,1}, \ldots, p_{m,1};$ $P_2 = p_{1,2}, p_{2,2}, \ldots, p_{m,2}\}$    and    $MDT_Q:\{Q_1 = q_{1,1}, q_{2,1}, \ldots, q_{n,1}; Q_2 = q_{1,2}, q_{2,2}, \ldots, q_{n,2}\}$ ($m = n$ because the length of the *MUD* record is the same for all building objects). In determining the DTW distances of $MDT_P$ and $MDT_Q$, the first step is to construct a distance matrix grid D of $m \times n$ elements. The values of each element ($d_{ij}$) in this matrix can be calculated as follows:

$$d_{ij} = \sqrt{\left(p_{i,1} - q_{j,1}\right)^2 + \left(p_{i,2} - q_{j,2}\right)^2} \tag{5}$$

where $p_{i,1}$ denotes the $MUD_w$ value at hour $i$ in $P$, $p_{i,2}$ denotes the $MUD_r$ value at hour $i$ in $P$, $q_{i,1}$ denotes the $MUD_w$ value at hour $i$ in $Q$, and $q_{i,2}$ denotes the $MUD_r$ value at hour $i$ in $Q$; and $d_{ij}$ represents the *MUD* difference between these two points. The algorithm can be reduced to finding a path through a number of grid points in this matrix grid, i.e., the warping path (*W*), with the kth element of *W* defined as $w_k = (i, j)_k$, representing the mapping of $P$ to $Q$. Thus, *W* can be expressed as:

$$W = w_1, w_2, \ldots, w_k, \ldots, w_K \qquad \max(m, n) \leq K \leq m + n - 1 \tag{6}$$

The regularization path (*W*) needs to satisfy the following constraints:

The boundary conditions require that the paths start at the element in the lower left corner of the matrix and end at the element in the upper right of the matrix, $d_{11}$ and $d_{mn}$, respectively.

The continuity condition, also known as the step condition, specifies that $w_k$ should be located in adjacent elements (including those in diagonal positions) of the matrix $w_{k-1}$.

The monotonicity condition restricts $w_k$ to be monotonically spaced in time, i.e., for $w_{k-1} = (i_{k-1}, j_{k-1})$, $w_k = (i_k, j_k)$ needs to satisfy $i_k \geq i_{k-1}$ and $j_k \geq j_{k-1}$.

There can be an exponential number of paths satisfying the constraints above, where there exists an optimal path with minimum accumulation:

$$\min \frac{\sum d_{ij}}{K} \tag{7}$$

The above equation can be minimized by the following recursive equation:

$$d_{cum,ij} = d_{ij} + \min\left\{d_{cum,i-1j-1}, d_{cum,i-1j}, d_{cum,ij-1}\right\} \tag{8}$$

where $d_{cum,ij}$ is the sum of the current $d_{ij}$ and the minimum of the cumulative distances of the previous elements. The obtained $d_{cum,ij}$ represents the Ndim-DTW distance between $P$

and $Q$. More information on computing multidimensional DTW distances can be found in the literature [35].

### 3.3. SOM Network

Compared to traditional clustering methods (e.g., k-medoids), neural networks can obtain a low-dimensional representation of each data point for network learning, and the low-dimensional representation learned is more suitable for clustering. In this study, a self-organizing map (SOM) network [54] is used to cluster building objects with similar time series. While general neural networks are trained based on the backward transfer of loss functions, the SOM method uses a competitive learning strategy that relies on neurons competing with each other to gradually optimize the network. The topology of the input space is maintained using the neighborhood function and is mapped to adjacent output neurons [54]. The SOM network structure has an input layer and an output layer (also called the competition layer). The number of neurons in the input layer is determined by the dimensionality of the input vector, and one neuron corresponds to one feature. The number of SOM neurons in the competitive layer determines the granularity and scale of the final model, which determines the accuracy and generalizability of the model.

Usage of the SOM method based on the Ndim-DTW distance involves the following steps:

Determine the number of competing layer neurons n;

At initialization, n samples from all building objects are selected as the initial nodes in the competition layer;

Select a random input sample $X_i$;

Iterate through each node in the competitive layer: calculate the Ndim-DTW distance between $X_i$ and the node. Select the node with the smallest distance as the winner node or best matching unit (BMU);

The nodes contained in the winning neighborhood (i.e., the neighborhood range of the winning node) are determined based on the neighborhood radius σ. Their respective update magnitudes are calculated by the neighborhood function, where the closer to the superior node, the larger the update magnitude, and the further away from the superior node, the smaller the update magnitude;

Update the weights of the nodes in the winning neighborhood,

$$W_{v(s+1)} = W_{v(s)} + \theta(u, v, s) \cdot \alpha(s) \cdot \left( D(t) - W_{v(s)} \right) \tag{9}$$

where $\theta(u, v, s)$ is a constraint on updates, i.e., the update magnitude factor; $W_{v(s)}$ is the current weight of node v; $\alpha(s)$ is the learning rate; and $D(t)$ is the average quantization error, i.e., the average distance from the nodes in the neighborhood to the winning node;

Complete one round of iterations (iteration number + 1) and return to step (3) until $D(t)$ meets the set value or reaches a certain number of iterations;

The bubble nearest neighbor function is chosen for the neighborhood function. It indicates that the update coefficients are the same as long as the neurons are in the superior neighborhood. The learning rate and neighborhood range in the SOM network decay with the number of iterations, and the decay function is $\frac{1}{1+t/T}$, and $\sigma_{t+1} = \frac{\sigma_t}{1+t/T}$, $\alpha_{t+1} = \frac{\alpha_t}{1+t/T}$, where $t$ represents the number of current iterations and $T$ represents half of the total number of iterations.

### 3.4. Initialization on the OC Algorithm

Similar to k-means methods, the correct initialization of competing layer neurons will affect the performance of SOM neural networks. Traditional methods based on random initialization are not effective in generating representative neurons for large datasets, and the results are often locally optimal. The data volume in this study is large, with a total of 122,544 building analysis units; thus, a modified O(logk)-Competitive (OC) algorithm [55] is used instead of random initialization to improve the clustering quality and accelerate the

convergence speed. The OC algorithm requires a randomly selected point from the dataset as the initial reference node. First, the Ndim-DTW distance between each sample $X_i$ and the existing nodes is calculated and expressed by $D(x)$. Then, the probability $P(x) = \frac{D(x)}{\sum D(x)^2}$ of each sample being selected as the next cluster centroid is calculated, after which the next initial node is selected by the roulette wheel method. The above steps are repeated until n neurons are selected.

### 3.5. Urban Function Identification and Hotspot Detection

The reconstructed POI dataset was used to identify the functions of the clustering results in Section 3.4. The enrichment factor (EF) [56] was used to characterize the relative richness of POI data at each building level:

$$F_{i,l} = \frac{n_{i,l}/n_i}{N_l/N} \tag{10}$$

where $F_{i,l}$ indicates the enrichment of building $i$ in class $l$ POIs; $n_{i,l}$ is the number of class $l$ POIs near the location of building $i$ (e.g., 10 m radius); $N_l$ indicates the total number of class l POIs; $n_i$ is the number of all POIs near the location of building $i$; and $N$ is the total number of POIs in the entire study area. A value of 1 for $F_{i,l}$ indicates that the enrichment level of class $l$ POIs is equal to the average level of the region, and $F_{i,l} > 1$ (or $< 1$) indicates that the enrichment of class $l$ POIs is greater (or lower) than the average value of the region. The average of the EF of all buildings within each cluster is used as the EF value of that cluster.

To reveal the macrostructure of urban functional areas, the relative richness of clusters can be further aggregated to the TAZ level. The Shannon index was used to describe functional diversity at the TAZ level:

$$H_i = -\sum_k p_k \ln(p_k) \tag{11}$$

where $H_i$ is the functional diversity of the $i$th street and $p_k$ is the proportion of buildings with function $k$. The Getis-Ord $G_i^*$ statistic is used to portray the analysis of functional hotspots in the city, and the areas with $G_i^*$ values greater than 2.58 are called functional hotspots [57].

$$G_i^* = \frac{\sum_{j=1}^n w_{i,j} H_j - \overline{H} \sum_{j=1}^n w_{i,j}}{S \sqrt{\frac{\left[n \sum_{j=1}^n w_{i,j}^2 - \left(\sum_{j=1}^n w_{i,j}\right)^2\right]}{n-1}}} \tag{12}$$

where $H_j$ is the Shannon index of street $j$, $w_{i,j}$ is the spatial weight between streets $i$, and $j$, $n$ is the total number of streets. Additionally,

$$\overline{H} = \frac{\sum_{j=1}^n H_j}{n} \tag{13}$$

$$S = \sqrt{\frac{\sum_{j=1}^n H_j^2}{n} - \left(\overline{H}\right)^2} \tag{14}$$

### 3.6. Accuracy Assessment

Since there are multiple mixes of building-level functional types, we cannot know the specific mixes before the SOM clustering results and cannot obtain validation samples in advance. Therefore, based on the method proposed by [58,59], a series of composites with different spatial scales was randomly selected in the study area, and the identification results were compared with ground truth survey data and Baidu Street View maps. To select sufficient validation samples, we chose spatial scale ranges of 500 × 500 m and 1000 × 1000 m.

## 4. Results

### 4.1. U-Matrix and Winner Matrix

We input 122,544 two-dimensional time series into the SOM network for clustering. This study set the initial number of neural nodes to 16 and initialize it with the OC algorithm to obtain 16 building objects as the initial neural nodes, i.e., 16 change patterns or function types can be captured. On this basis, a DTW-SOM net with a 4 × 4 rectangular grid layout was constructed. The initialized learning rate and neighborhood radius were set to 0.5 and 0.3, respectively, and the average quantization error was set to 0.05 for iterative training. Among them, the learning rate and neighborhood radius were used to control the operation of the network, and the average quantization error was used to end the iteration. The model was trained to obtain the U-Matrix and winner matrix (Figure 4). Each grid in the U-Matrix represents a winning neuron (BMU), and their indices were labeled with numbers. The magnitude of the value in the U-Matrix indicates the similarity between the BMU and its neighboring winning neurons in the input space (i.e., the Ndim-DTW distance) and was encoded using colors, with larger values being darker. Figure 2a shows that BMUs 1 and 2 are adjacent and have less difference in the labeled values; thus, they are more likely to be combined into one category. BMU 16 is more different from the surrounding BMUs and is likely to be divided into separate categories. The winner matrix (Figure 5b) indicates the number of buildings contained in each BMU category, with the highest number of buildings in BMU4 being 20,071 and the lowest number of buildings in BMU16 being 181.

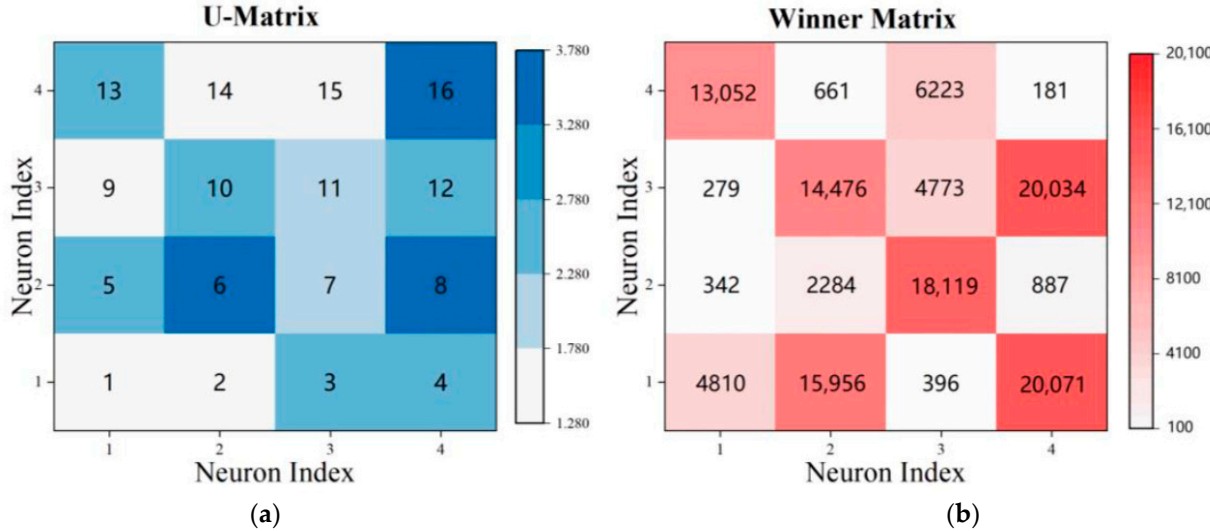

(**a**)                                                  (**b**)

**Figure 5.** (**a**) U-Matrix: The average of the Ndim-DTW distance between the BMU and the neighboring BMUs. A smaller distance means that it is more likely to be grouped with the neighboring BMUs; (**b**) Winner Matrix: The number of buildings contained in each BMU category.

### 4.2. Change Curve in Each BMU

We set 16 neurons to compete and finally obtain a sequence of 16 BMUs (Figure 6). Each BMU included two characteristic change curves of the working population (red color) and residential population (blue color). Although each BMU represents a change pattern, some BMUs, such as BMU1 and BMU2, have similar change curves. On weekdays, the working population starts to decrease at approximately 07:00 and starts to increase at 17:00, finally reaching stability at approximately 21:00. On weekends, the timing and magnitude of this change are smaller. There is no significant difference in the number of residents on weekdays and weekends compared to the change in the number of working people. In addition, BMU5 and BMU9 have similar patterns of change (Figure 6).

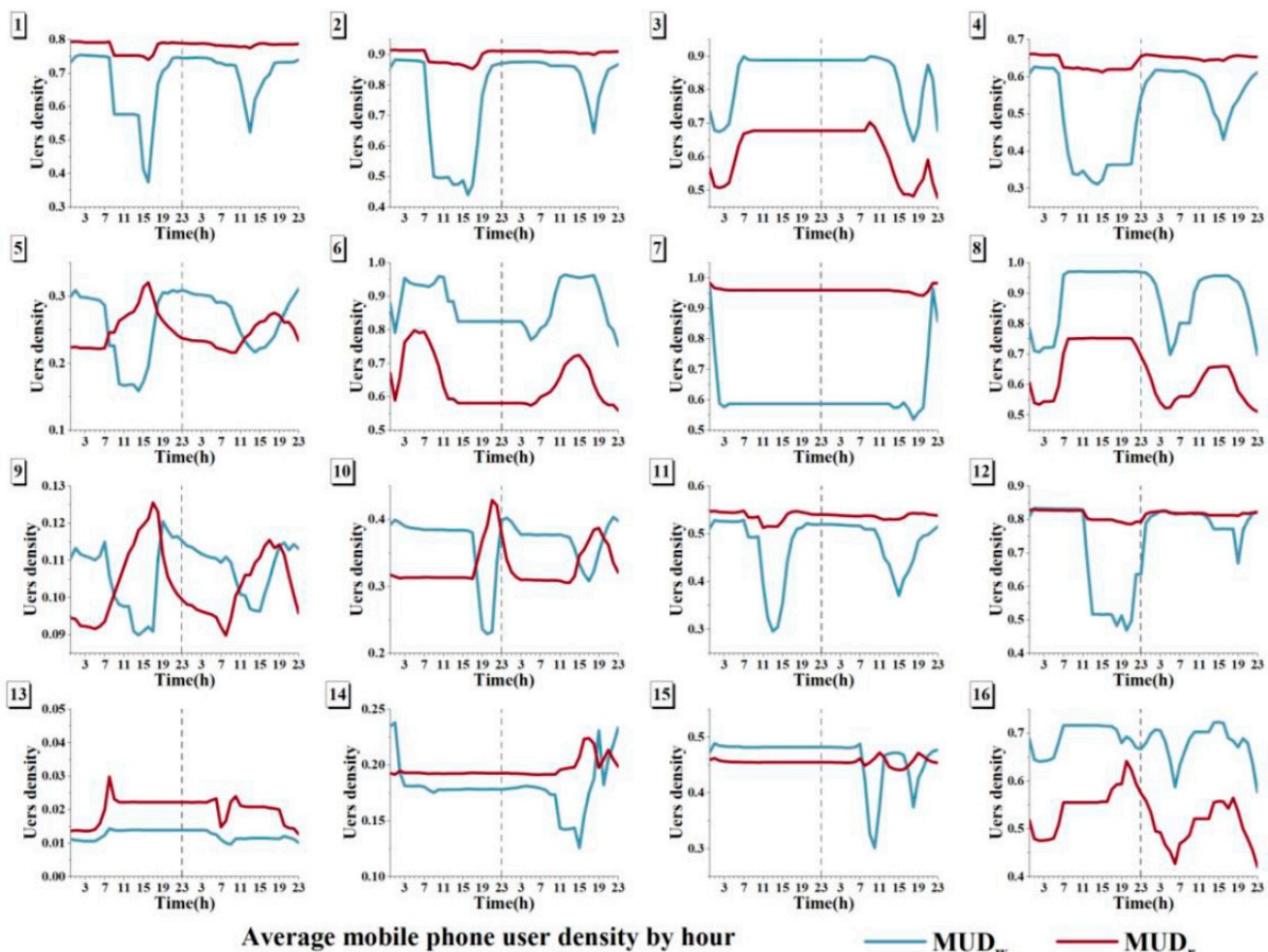

**Figure 6.** Time series change curves in each BMU. A total of 16 BMU time series were obtained, where the red curve represents the residential population and the blue curve represents the working population. The *y*-axis represents the population density. The *x*-axis represents the temporal variation in both weekday and weekend patterns, where the left side of the dashed line represents the weekday pattern and the right side of the curve represents the weekend pattern.

### 4.3. Enrichment Factors

To further understand the functional type of each BMU model, the enrichment factor (EF) of each building was calculated based on the reconstructed POI categories, and in turn, the average EF of each BMU cluster was obtained (Table 3). The larger the POI enrichment factor in the BMU cluster is, the higher the POI enrichment of the type contained in the cluster. For example, in BMU1, only the POI enrichment factor of residential type was greater than 1, while all other types were less than 1, indicating that the cluster is mainly residential in function. In BMU5, the POI enrichment factors for both residential and industrial types were greater than 1, indicating that the cluster is dominated by a mixture of both residential and industrial functions. Since the object of this paper is the function of each building, the enrichment factor of the parks and greenspace type (PG) was not considered.

**Table 3.** Enrichment factors for different categories of POIs grouped by BMU clusters.

| BMU | IF | AD | MD | ED | SC | BS | SM | RC | PG |
|-----|-----|-----|-----|-----|-----|-----|-----|-----|-----|
| 1 | 0.63 | 0.22 | 0.86 | 0.94 | 0.89 | 0.68 | 0.62 | **1.59** | 2.94 |
| 2 | 0.25 | 0.02 | 0.98 | 0.83 | 0.01 | 0.15 | 0.65 | **1.75** | 0 |
| 3 | 0.02 | **1.65** | **1.92** | 0.52 | **1.61** | **1.95** | 1.05 | 0.73 | 1.81 |
| 4 | 0.72 | 1.23 | 0.24 | 0.56 | 0.64 | 0.53 | 0.6 | **1.64** | 0.53 |
| 5 | **1** | 0.34 | 0.86 | 0.5 | 0.68 | 0.57 | 0.44 | **1.88** | 1.23 |
| 6 | 0.2 | **1.54** | **2.22** | 0.68 | 1.1 | **1.43** | 0.99 | 0.87 | 2.27 |
| 7 | 0.37 | 0.37 | 0.3 | 0.99 | 0.11 | 0.25 | **1.84** | **1.45** | 0.31 |
| 8 | 0.32 | 0.89 | 0.9 | 0.41 | 1.47 | **3.1** | 1.22 | 0.55 | 1.57 |
| 9 | **2.56** | 0.94 | 0.62 | 1.35 | 0.68 | 0.71 | 0.4 | **1.95** | 0.74 |
| 10 | **1.96** | 1.22 | 1 | 1.1 | 0.56 | 0.64 | 0.46 | **1.87** | 0.98 |
| 11 | 0.66 | 0.39 | 0.81 | 0.86 | 0.94 | 0.69 | **1.63** | **1.57** | 1.38 |
| 12 | 0 | 0.31 | 0.92 | 0.76 | 0 | 0.22 | 0.67 | **1.59** | 3.18 |
| 13 | **2.43** | 0.81 | 0.69 | 3.2 | 1.16 | 0.8 | 0.59 | 1.15 | 0.61 |
| 14 | 0.63 | 0.65 | 0.69 | **1.52** | 0.67 | 0.68 | 0.4 | **1.96** | 0.36 |
| 15 | 0.91 | 0.71 | 0.92 | **1.32** | 1.12 | 0.58 | 0.52 | **1.72** | 3.34 |
| 16 | 0.17 | **1.88** | **1.9** | **2.13** | **2.24** | 1.12 | 0.94 | 0.8 | 5.8 |
| 16 | 0.17 | 1.88 | 1.9 | 2.13 | 2.24 | 1.12 | 0.94 | 0.8 | 5.8 |

Note: IF = industrial facilities; AD = administrative; MD = medical; ED = educational; SC = sports and cultural; BS = business; SM = shopping malls; RC = residential communities; PG = parks and greenspace.

### 4.4. Building-Level Urban Function Types

We selected the data in each BMU cluster that were two times higher than the average value and had an enrichment factor value greater than 1 (Table 3 in bold). The corresponding POI categories were used to characterize the BMU functions, and the final 16 BMU patterns were further aggregated into nine functional clusters (Table 4), namely: residential; business/shopping malls/social; residential/industrial; business/social; shopping malls/residential; business; industrial; educational/residential; and social/educational. Among them, the residential cluster contains four BMU patterns, namely, BMU1, BMU2, BMU4 and BMU12; the residential/industrial cluster contains three patterns, BMU5, BMU9 and BMU10; the shopping malls/residential cluster contains two patterns, BMU7 and BMU10; the educational/residential cluster contains BMU14 and BMU15; and the business/shopping malls/social, business/social, business, industrial and social/educational clusters contain only one BMU pattern.

**Table 4.** Functional clusters and types to which the 16 BMUs belong.

| Cluster | BMU | Function |
|---------|-----|----------|
| 1 | 1,2,4,12 | Residential |
| 2 | 3 | Business/shopping malls/social |
| 3 | 5,9,10 | Residential/industrial |
| 4 | 6 | Business/social |
| 5 | 7,11 | Shopping malls/residential |
| 6 | 8 | Business |
| 7 | 13 | Industrial |
| 8 | 14,15 | Educational/residential |
| 9 | 16 | Social/educational |

Figure 7a shows the nine functional area classification results overlaid with the FROM-GLC10 land cover types. Although there are only three types of single-functional buildings, namely, residential, industrial and business buildings, and the remaining six are all mixed types, single-functional buildings account for 63.75% of the total number. Among them, the number of residential buildings is 51,685, accounting for 43.4% of the total number, followed by industrial buildings, accounting for 18.3% of the total number (Figure 7b).

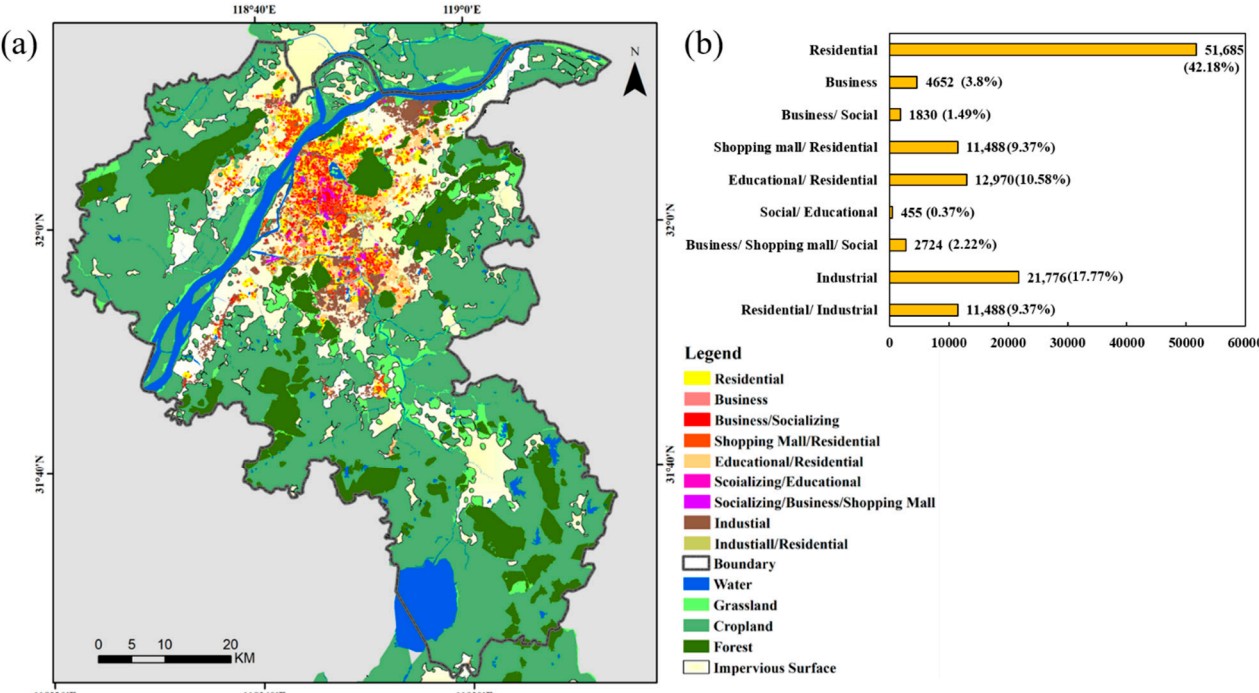

**Figure 7.** (**a**) Spatial distribution of functional clusters, with the land cover map in the background. (**b**) The number of buildings in each functional area.

### 4.5. TAZ-Level Urban Functional Hotspots

In this study, the functional diversity of each TAZ was calculated using the Shannon index, and a total of seven "hotspots" were identified based on the TAZ-level functional diversity using hotspot analysis. Figure 8a,e are the two university citys in Nanjing, Xianlin University City (XL) and Jiangning University City (JN), respectively, where XL has eight functional types and JN has seven functional types, both with the highest number of educational and residential buildings. Figure 8b shows the Qiaobei business district (QB), which has several large shopping malls and residential districts and contains nine functional types, with a high proportion of commercial and residential buildings. Figure 8c shows Xinjiekou (XJK), which is the CBD of Nanjing's oldest commercial center, with perfect supporting facilities, eight functional types, and a high proportion of business and commercial buildings. Figure 8d is the Nanjing South Railway Station (NSR), surrounded by several industrial parks and shopping malls, with eight functional types, among which residential is the most frequent and residential/industrial is the second most frequent. Figure 8f shows the Baima Road Block (BRB), known as Nanjing's livable block; it is dominated by residential buildings with mature community development and is surrounded by well-developed educational and living facilities with six functional types. Figure 8g shows the Hexi CBD (HX), a modern service industry concentration area integrating business centers, industrial parks and residential areas with complete supporting facilities. Among these seven hotspots, XL has the widest spatial extent (area of 37.74 km$^2$), while XJK, BRB and HX have smaller areas (3.48 km$^2$, 4.48 km$^2$ and 3.19 km$^2$, respectively). However, XJK (Figure 8c) has a more balanced number of buildings of all types, with the highest Shannon diversity index (highest column in the figure). The Whitehorse Road hotspot (Figure 8f) is dominated by residential buildings, with a lower percentage of other types, and therefore, it has a lower diversity index value and the lowest column height. However, XJK (Figure 8c) has a more balanced number of buildings of each type, and it has the highest Shannon diversity index (highest column in the figure). The BRB (Figure 8f) is dominated by residential buildings, with a lower percentage of other types, and therefore, it has a lower diversity index value (lowest column height).

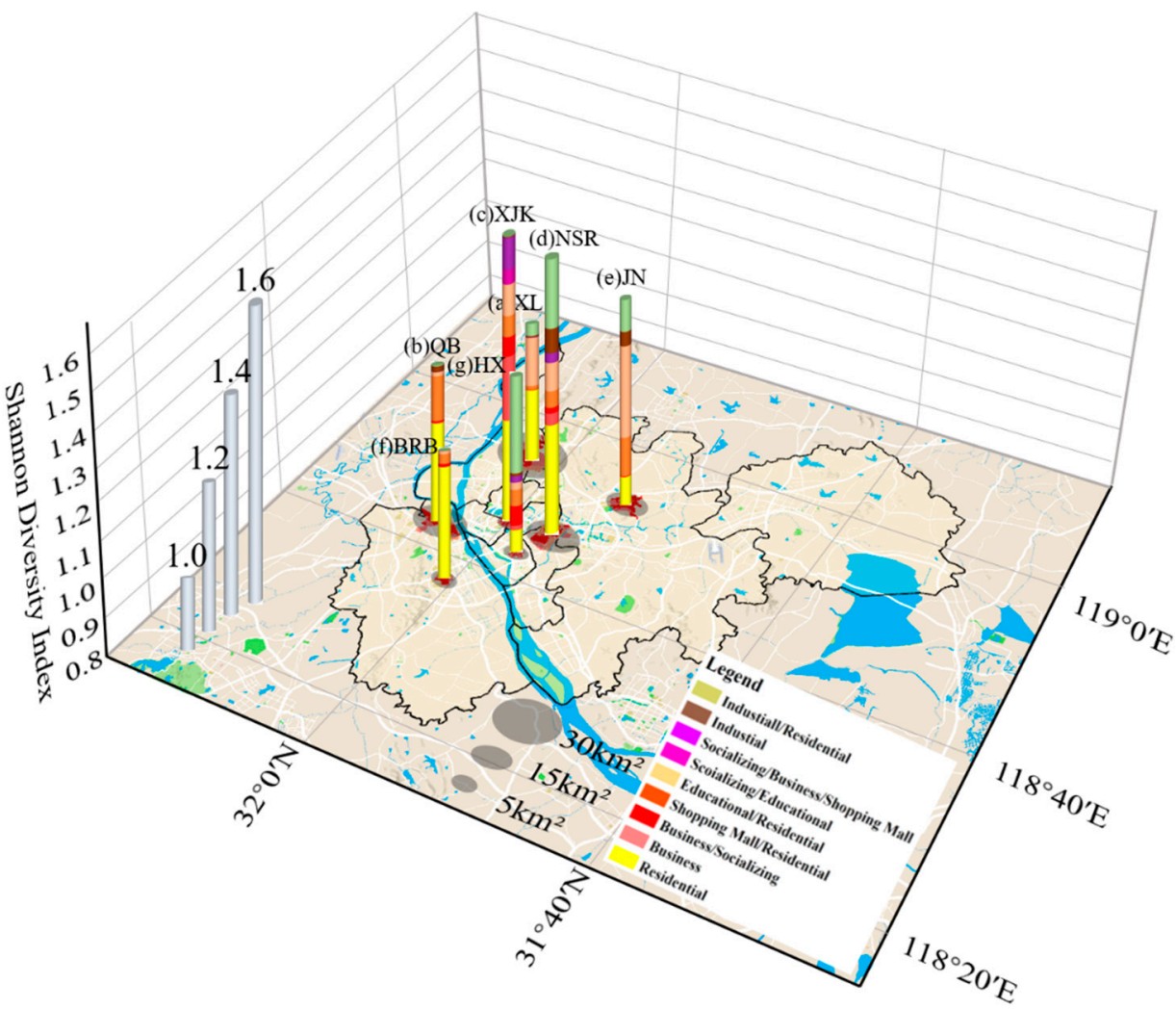

**Figure 8.** The location and size of hotspot areas. The *x*-axis and *y*-axis represent longitude and latitude, respectively; the *z*-axis represents the average Shannon index of each hotspot area. (**a**) Xianlin University City, (**b**) Qiaobei business district, (**c**) Xinjiekou commercial and shopping district, (**d**) Nanjing South Railway Station, (**e**) Jiangning University City, (**f**) Bema Road Block, and (**g**) Hexi CBD. Each column contains colors of different proportions, representing the proportion of the number of buildings in different clusters. The red areas under the columns are the actual extent of the hotspots, and the gray circles indicate the size of the hotspots.

### 4.6. Accuracy Assessment

A validation sample of 186 single-function and 67 mixed-function buildings was selected on a spatial scale of 500 × 500 m. Among them, 16 single-function and 9 mixed-function buildings were predicted incorrectly; thus, the accuracy rates were 91.4% and 86.57%, respectively, and the overall accuracy rate was 88.99% (Table 5). At the spatial scale of 1000 × 1000 m, 159 and 462 single- and mixed-function buildings were selected for the validation sample, and the prediction accuracy rates were 91.82% and 85.28%, respectively, with an overall prediction accuracy rate of 88.55% (Table 5). The accuracy rate for urban functional areas is always above 85% at different spatial scales, which proves the robustness of our model. Overall, the accuracy rates for single-function buildings at different scales are all above 91%, which is significantly higher than those for mixed-function buildings.

**Table 5.** Comparison of the predicted results with ground truth survey data at different scale ranges.

| Site | Size | Single | | | Mixed | | | C |
|---|---|---|---|---|---|---|---|---|
| | | N1 | N2 | C1 | N3 | N4 | C2 | |
|  | 500 m × 500 m | 186 | 16 | 91.4% | 67 | 9 | 86.57% | 88.99% |
|  | 1000 m × 1000 m | 159 | 13 | 91.82% | 462 | 68 | 85.28% | 88.55% |

Note: N1, N2, N3 and N4 are the number of single-function buildings, the number of incorrectly predicted single-function buildings, the number of mixed-function buildings and the number of incorrectly predicted mixed-function buildings, respectively. C1 and C2 are the accuracy rate for the inferred single-function buildings and the accuracy rate for the inferred mixed-function buildings, respectively. c represents the overall accuracy rate.

## 5. Discussion

### 5.1. Comparison of Different Classification Methods

To explore the attribute information of the mobile user data and whether the SOM method based on the Ndim-DTW distance proposed in this paper is effective, the results of this paper are compared with those of the traditional DTW distance-based K-medoids method [34]. When the mobile user data are used for the DTW similarity calculation without considering the attribute information, only the user density data ($MUD_m$) are used as a measure of population size, and the classification results show an average accuracy of 76.3% (Table 6). Second, if the user density data with attributes ($MUD_r$, $MUD_w$) are used for the calculation of the Ndim-DTW distance metric, the K-medoids clustering accuracy is improved to 80.9%, and it is seen that the clustering accuracy is improved by 4.6% after adding the population attribute features. If only one-dimensional $MUD_m$ data without population attributes are used but the SOM clustering method proposed in this paper is used, the clustering accuracy is improved to 81.4%. This result shows that the clustering method contributes more to the clustering results than the dimension of population data attributes. The reason is that after the K-medoids method finds the most similar class for each input data, only the parameters of this class are updated. The SOM method, on the other hand, updates the adjacent nodes, which are less affected by noise data than K-medoids [54]. If both $MUD_r$ and $MUD_w$ two-dimensional attribute data and the SOM clustering method are used, the results show the highest classification accuracy, 88.7% (Table 6). It can be seen that the mobile subscriber data with attribute information and the SOM method based on the Ndim-DTW used in this paper can effectively improve the discriminative accuracy of individual building functional categories.

**Table 6.** Accuracy comparison of different data and method combinations.

| Methods | Data | Average Accuracy (%) |
|---|---|---|
| DTW + K-medoids | $MUD_m$ | 76.3% |
| Ndim-DTW + K-medoids | $MUD_r$, $MUD_w$ | 80.9% |
| DTW + SOM | $MUD_m$ | 81.4% |
| Ndim-DTW + SOM | $MUD_r$, $MUD_w$ | 88.7% |

### 5.2. Characteristics of Urban Hotspots

(Chen and Liu et al., 2017) identified five "hotspots" in the Yuexiu District, Guangzhou, based on TAZ-level functional diversity and concluded that the identification of these hotspots solves, to a certain extent, the problem of defining "centers" or "sub-centers" [34]. The same method was used to identify nine functional hotspots in Nanjing (Figure 8), but

further evidence is needed to determine whether these areas are indeed the "center" or "sub-center" of Nanjing.

Urban hotspots contain a rich variety of functional types within them, and these different functional areas are often initially defined by urban planning. For this purpose, we compared the Nanjing 2018–2035 urban planning map (http://ghj.nanjing.gov.cn/ghbz/ztgh/201705/t20170509_874089.html, accessed on 18 November 2021) with the functional types of the seven detected functional area hotspots and their distribution (Figure 9). As the CBD of Nanjing, Xinjiekou was planned as a commercial area at the beginning of Nanjing's construction and then gradually developed into a commercial and financial center. It can be seen from the planning map that commercial, business and residential buildings occupy the main part, which is consistent with our results (Figure 9c). The Hexi CBD and Nanjing South CBD are also urban financial centers that were built through planning efforts, where business, industrial and residential buildings are the most numerous (Figure 9d,g), which is basically consistent with the planning map (Figure 9d,g). However, there are often multiple mixed types of buildings actually distributed in a single type of planning parcel (Figure 9f,g). This may be related to two reasons: first, our classification results are based on individual buildings, which are more refined, and traditional planning maps are coarser in terms of streets or TAZs, resulting in scale differences [49,60]; second, functional areas are subject to change by the long-term activities of the population, resulting in clusters of different functional buildings on the original land plan type [15,58]. Each of these hotspot areas has different functional characteristics influenced by urban planning: commercial hotspots (XJK and QB), business hotspots (HX and NSR), educational hotspots (XL and JN), and residential hotspots (BRB). The type of hotspot is determined by the building functions contained within the hotspot or the type of land use of the parcel (Figure 9). In the 2020 version of the Nanjing city plan (http://ghj.nanjing.gov.cn/ghbz/ztgh/201705/t20170509_874089.html, accessed on 18 November 2021), the central city of Nanjing includes "one main", namely, Jiangnan main city, and "new" is the new main city of Jiangbei. Compared with the previous version of the city master plan, the status of the Jiangbei area has been greatly enhanced, and the development of embracing the river has become one of the most important spatial development strategies for Nanjing (Nanjing City Master Plan (2018–2035) draft). In terms of the spatial distribution of hotspots, the Jiangbei New Area has the BRB and QB hotspots to meet the living and residential requirements of the population, which is basically consistent with the requirements of urban development. In addition, the educational hotspots (JN and XL) are located in the northern and southern areas of the city, which are both within the planned "National Science Center Demonstration Zone" (Nanjing Urban Master Plan (2018–2035) draft). The spatial distribution of the different types of hotspots that we identified is consistent with the overall layout of Nanjing city planning, thus proving the effectiveness of our method.

If urban planning forges the basic outline of the city's functional areas, then the final formation of functional areas is closely related to the actual needs and usage of residents [5]. Urban populations tend to cluster toward urban centers, making urban centers tend to have high population densities [11,61]. To investigate whether the functional hotspots that we identified have a high population density, we counted the average number of people working and living per hour in each hotspot area separately in a week, while three non-hotspot areas were selected as controls, namely, the Bajia Lake Business District (BJL), Zhangcun Industrial Park (ZC) and Changjiangzhijia District (CJ) (Figure 10a). The BJL is a typical commercial/residential area with mainly shopping malls and residential buildings; the ZC is a business/industrial area with a large number of business offices and factory buildings; and the CJ is a residential area with mainly residential buildings. In the hotspot areas, both attribute population densities are high (Figure 10b,c). In contrast, in non-hotspot areas, both population densities are lower or cannot be at high values at the same time. For example, in the BJL, both attribute population densities are lower than hotspot areas. In the ZC, the working population density is higher than some hotspot areas (Figure 10b), but its residential population density is the lowest (Figure 10c). The same is true for the

CJ, which has a high residential population density and a low working population density (Figure 10b,c). It can be seen that hotspot areas not only have a variety of urban function types but also are high-density population distribution areas. In addition, the analysis of the two types of population densities reveals that there is a problem in capturing the pattern of population activity across functions by only looking at the change in population density for a single attribute: the high and low values of the two attributes "cancel out" each other, thus "smoothing out" the difference in patterns across functions [62]. For example, the CJ, with a high residential population density and a low working population density, and the ZC, with a high working population density and a low residential population density, represent two different functional types, but they may have similar total population densities. The population attribute information helps to improve the accuracy of the fine-grained classification of functional areas (Table 6). In this study, we used only two attributes, work and residence, and we can try to consider more population attributes in the future, such as the difference in the activity space and pattern between different age groups and genders.

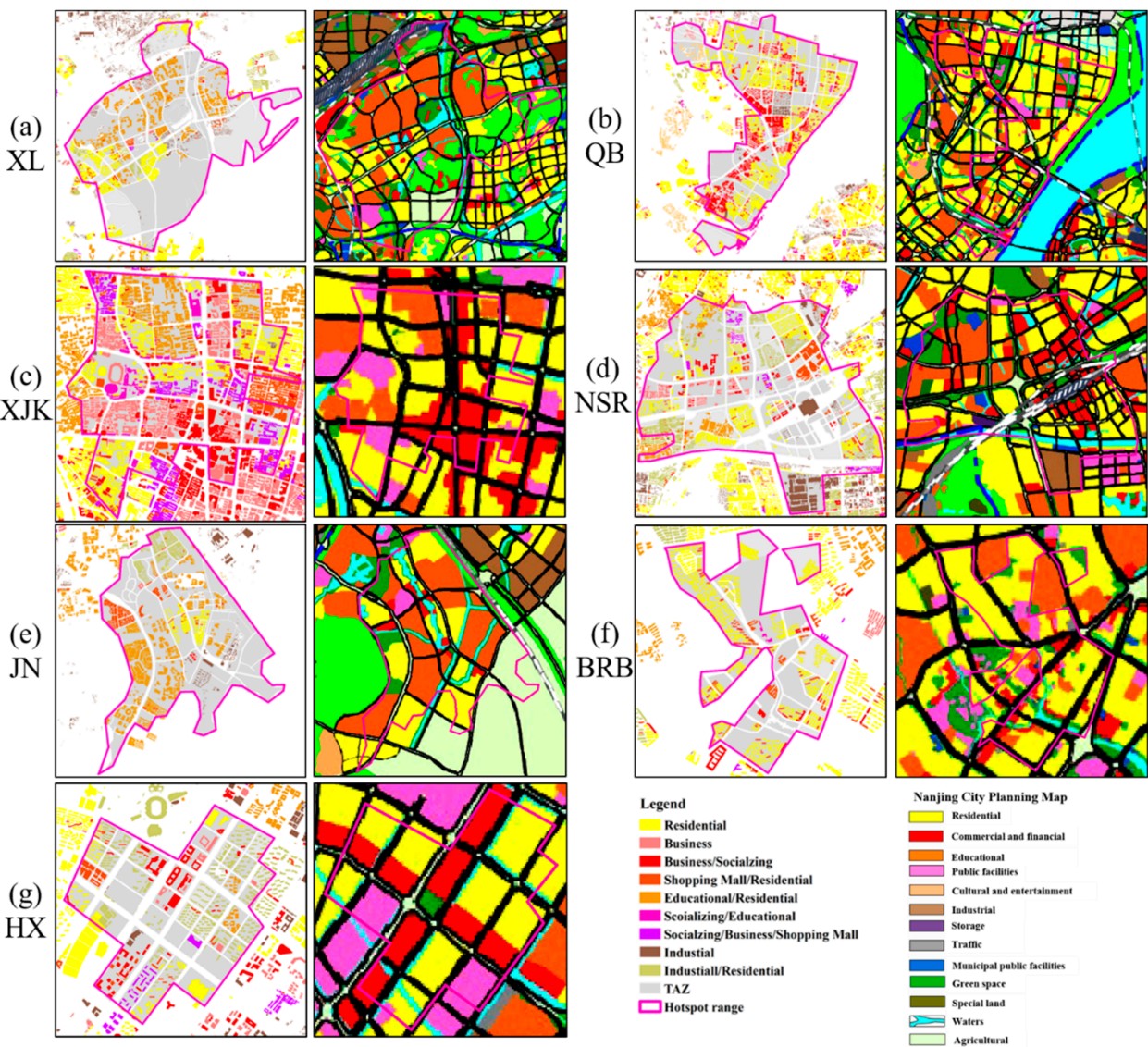

**Figure 9.** Comparison of functional zoning results within the hotspots with the city planning map.

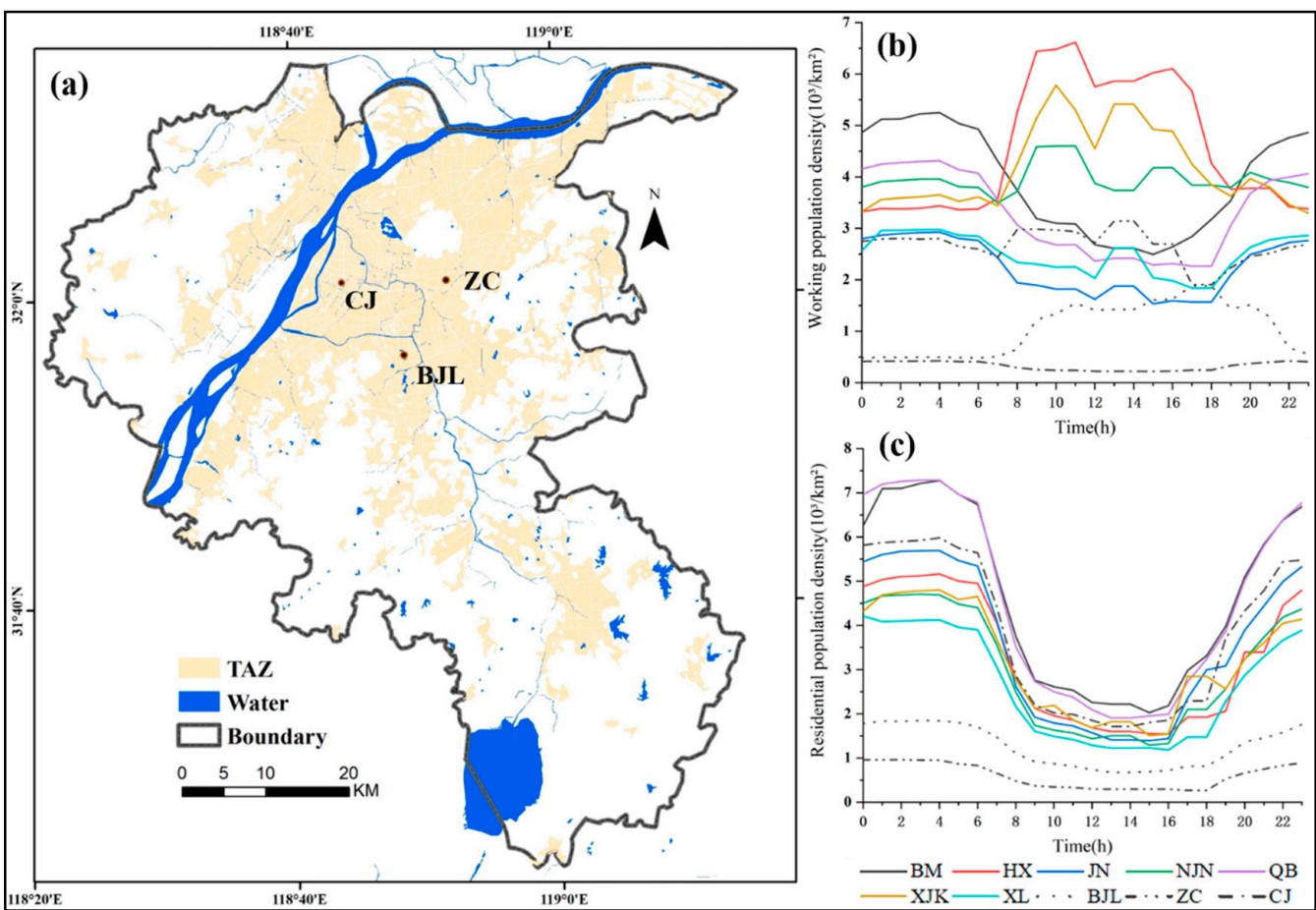

**Figure 10.** (**a**) Spatial distribution of non-hotspot areas. (**b**) Time series of the working population density of hotspots and non-hotspots. (**c**) Time series of the residential population density of hotspots and non-hotspots.

Although these hotspots always exhibit a high crowd density, we cannot infer whether the hotspots are formed because of the high crowd density or because the functional areas have a wide range of categories that attract crowds to gather. From the results of this study, it is clear that crowd density influences the formation of hotspots. For example, the two university towns (Figure 8a,e) are among the hotspots identified in this study, which are different from the traditional urban centers, probably due to the special time of cell phone data collection. The mobile subscriber data collection period used in this study was from 18–24 February 2019, which is the time when Nanjing colleges and universities started school one after another. We collected information on the start time of colleges and universities in spring 2019 and found that 12 colleges and universities in Nanjing started school one after another during this time (Table 7). Crowds are heavily concentrated in schools and station places during this time, thus making these areas hotspots in a short period of time. This finding also provides an important insight: the spatial structure of urban functional areas will change dynamically due to crowd activities. (Tu and Cao et al., 2017) discerned urban functional areas based on crowd activity information inferred from cell phone location and social media data and found that many urban areas provide different functions depending on the type and extent of human activities, i.e., urban functions are dynamically changing [19]. Our study also captures the structural changes in urban functions that occur at particular time periods.

**Table 7.** Opening times of some universities in Nanjing in 2019.

| Name | Opening Time |
|---|---|
| Nanjing Audit University; Nanjing Medical University; Nanjing University of Posts and Telecommunications; Nanjing University of Chinese Medicine | 17 February 2019 |
| Southeast University Chengxian College; Nanjing University of Finance & Economics | 22 February 2019 |
| Nanjing Normal University | 23 February 2019 |
| Hohai University; Nanjing University; Nanjing University of Aeronautics and Astronautics; Nanjing University of Science and Technology; China Pharmaceutical University | 24 February 2019 |

## 6. Conclusions

In this study, an SOM neural network method based on the Ndim-DTW distance was proposed to extract the functional categories of individual buildings in cities using the information of the work and residence attributes of mobile users. It was found that the SOM method and two-dimensional attribute data improved the accuracy of functional area identification by 8.1% and 7.3%, respectively, compared to using the traditional K-medoids clustering method and population density single-attribute mobile user data. In addition, analyzing population patterns by attribute can avoid the problem that the high and low values of different attributes "cancel out" each other and "smooth out" the differences in the patterns of different functions. This study found that there are nine types of functional categories of buildings in Nanjing and, with the exceptions of the residential, industrial and commercial categories, all of them are mixed categories. However, single-function buildings account for the majority (63.75%). In this paper, we used mobile-aggregated data for the week of 18–24 February 2019, to discover a total of seven hotspot areas in Nanjing, the formation of which is not only related to the diverse functional categories that the areas possess but also closely related to the distribution of high-density populations, which is characterized by spatial and temporal dynamics. As a result, our results can provide a scientific basis for long-term urban planning and certain functional requirements in special periods.

**Author Contributions:** Conceptualization, Zhenglin Song; methodology, Zhenglin Song and Hong Wang; software, Zhenglin Song; validation, Zhenglin Song; formal analysis, Zhenglin Song; investigation, Zhenglin Song, Shuhong Qin, Xiuneng Li, Pengyu Meng and Yicong Wang; resources, Zhenglin Song and Yi Yang; data curation, Zhenglin Song; writing—original draft preparation, Zhenglin Song; writing—review and editing, Zhenglin Song; visualization, Zhenglin Song; supervision, Hong Wang. All authors have read and agreed to the published version of the manuscript.

**Funding:** This research was funded by the National Natural Science Foundation of China (No. 41471419 and No. 31971579).

**Data Availability Statement:** Not applicable.

**Acknowledgments:** The authors would like to thank the editors and anonymous reviewers, whose detailed comments and suggestions have notably helped to improve the paper.

**Conflicts of Interest:** The authors declare no conflict of interest.

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
