# Peer review of "Building-Level Urban Functional Area Identification Based on Multi-Attribute Aggregated Data from Cell Phones—A Method Combining Multidimensional Time Series with a SOM Neural Network"

_ijgi, doi:10.3390/ijgi11020072_

Round 1

Reviewer 1 Report

Paper is very clear in presenting the goal of identifying urban funcionality at a very detailed level and in describing the methodology and results of the experiment. Introduction, which includes the background or literature review, is very complete and guides the reader to the topic. The materials and methods section is also very detailed, as well as the analytics presentation.

However, I think there are parts of the manuscript that turn the reading more difficult than necessary.

The first of them is the justification for the reconstruction/regeneration of POI data, where a reason for the adoption of the second method is not explained.

The practical implications of the work, in particular those numbered as #2 and #3, need to be more clearly explained. #2 is not directly linked to the work, while #3 could be enhanced through a clear example. Moreover, these paragraph, after Table 7, fits better in the Conclusions than in the current section it is in (Discussion).

Reviewer 2 Report

Dear authors, 

thank you for your paper and interesting research. I found the research done interesting and well organized, and the topic is surely worth an investigation also relying on the data and tools you correctly used. 

A set of points and questions were raised when reading your paper. 

  • literature review. It should be extended and kept more international. The issue of defining functional urban areas and, more broadly, gravitation areas of a city, is a classical question of urban geography and urban planning. Some references on the classical studies on such an issue should be cited. Authors cited are mainly locals and with little reference to the wider literature available on the issue. A non-exaustive list of references at the end of this box. 
  • Methods. Kernel Density Estimation is cited but no reference at all on the function, parameters and motivation for using it is done. Please improve and integrate such a section.
  • POIs' classification is an interesting point in the definition of the funcional areas, however little attention was put on the starting points for classification. Why such a categorization was done? Again, there is a huge load of literature and studies on such issues, and order on such point should be done in order to reinforce the replicability of the method used, particularly for planning issues.

ALONSO W., A Theory of the Urban Land Market, in “Papers and Proceedings of the Regional Science Association”, 6, 1960, pp. 149-157. 

BERRY, B. J. L., Geography of Market Centers and Retail Distributions, Printice-Hall, Englewood Cliff, NJ 1967.

Borruso G., Porceddu A. (2009) A Tale of Two Cities: Density Analysis of CBD on Two Midsize Urban Areas in Northeastern Italy. In: Murgante B., Borruso G., Lapucci A. (eds) Geocomputation and Urban Planning. Studies in Computational Intelligence, vol 176. Springer, Berlin, Heidelberg. https://doi.org/10.1007/978-3-540-89930-3_3

CAROL H., The hierarchy of central functions within the city, in “Annals of the Association of American Geographers”, 1960, pp. 419-438.

CHAINEY S., REID S. e STUART N., When is a hotspot a hotspot? A procedure for creating statistically robust hotspot maps of crime, in KIDNER D., HIGGS G. and WHITE S. (a cura di.), Socio-Economic Applications of Geographic
Information Science, Innovations in GIS 9, Taylor and Francis, 2002, pp.
21-36.

CUTHEBERT A. L., ANDERSON W. P., Using Spatial Statistics to Examine the Pattern of Urban Land Development in Halifax-Dartmouth, in “The Professional Geographer”, 54, 2002, 4, pp. 521-532.

GATRELL A., Density Estimation and the Visualisation of Point Patterns. In: HEARNSHAW H. M., UNWIN D. (a cura di), “Visualisation in Geographical Information Systems.” Chichester, Wiley. 1994, pp. 65-75.

HARRIS C. D., ULLMAN E. L., The Nature of Cities, 1945, 242, pp. 7-17. 

HOCH I., WADDEL P., Apartment Rents: Another Challenge to the Monocentric Model. In “Geographical Analysis” 25, 1993, 1, pp. 20-34

HOYT H., The Structure and Growth of Residential Neighborhoods in American Cities, U.S. Government Printing Office, Washington, D.C., 1939.

MURPHY R. E., VANCET J. E., Delimiting the CBD. Economic Geography 30(3):189-222, 1954.

WADDEL P., BERRY B. J. L., HOCH I., The Intersection of Space and Built Form. In “Geographical Analysis” 25, 1993, 1. pp. 5-19.

YEATS M. H., GARNER B. J., The North American City, New York, 1976.

Reviewer 3 Report

First of all, I would like to say that this manuscript focuses on a very interesting research problem.

TITLE

The article’s title is suitable with the content of the paper. However, The comparative analysis is welcome in line with both the text body and the main findings of the research.

ABSTRACT

The abstract is well-designed and briefly express the present research thus being of interests and readable thus capturing the reader’s attention. It present in an appropriate manner the main research hypothesis, the problem statement, the methods and the main findings.

KEY WORDS

The key words are appropriate to the present research and are clearly stated.

ORIGINALITY

The article meets a high level of originality argued by the main research theme and the research hypothesis. Furthermore, the originality of the paper is highlighted by the main results of the paper.

The authors construct a well-designed theoretical background closely related to the current specialised literature in the field. However, please extend the description of the functional areas and use more European sources.

A short recommendation I would like to made, it is stated in the final part of this review form.

THE PAPER S STRUCTURE

The objectives seem to be clear formulated as well as the investigation is drawn. The structure of the article does not comply with the journal's standards and requires that it meets the requirements of the publication taking into account the logic of the article. I suggest the classic layout of a scientific article. Please remove the subsections because in this system it looks like chapters of a monograph.

THE METHODS

The methodological design is appropriate and the methods fit well to the present investigation. The comparative analysis highlight well the main processes and characteristics in spatial planning in the rwo sampled case studies. The methods used in the study are well expressed both in the graphical form as well as in the main text of the manuscript.

However, complete the network-building content with their learning and validation process. Complete the techniques that were used in the parameterization of algotymes. No scientific discussion of network problems. How did you solve the research problem or is it a velo-grading task? Were the methods supervised or maybe partially supervised? Tested to binary or not? Whether the amount of data is small or large, or how much data the network needs, how much data is required. How did you validate the operation of the network? I am able to conclude how, but there is no description for a specialist urban planner without knowledge of artificial intelligence. Does the data tend to aggregate? Optimization and globalization methods have been used.

THE MAIN ANALYSIS

The main research is well design and appropriate conducted in line with the main questions in spatial planning.  I suggest the authors to be a little more specific.

CONCLUSIONS

The conclusions fit well summarising the main ideas of the present analysis.

THE GRAPHICAL SUPPORT

The graphical support is well formatted, appropriate illustrating the text content.

THE ENGLISH LANGUAGE

I think the English is ok as far as I could see. I enjoyed to read this paper in English and the language seems well but I think that an opinion of a native English speaker is welcomed. In other words, if the authors used a specialised proofreading services and they could prove this aspect I trust the opinion and the work of this proofread service. On the other hand, I put my trust regarding the English language on the journal editors but I repeat the language seems well.

RECOMMENDATIONS

Finally, I recommend the publication of this paper with some minor revision considering the above mentioned aspects, references and citations.

I want to see the revised version of this paper before publication for a final acceptance and to ensure that the revision has been completely and carefully made.

Round 2

Reviewer 2 Report

Please refer to the comments expressed after round 1 of revisions.

I do not feel my suggestions were met. 

In particular (as in my notes) there is the need to:

  • reinforce the references on international research;
  • provide a sounder formalization on the methodology adopted;
  • Insert POIs classification into a wider framework of the research on the topic. In simpler terms: how is the POIs choice done with reference to similar research on the topic? What categories of POIs are used in other similar research and/or indicators?

Previous evaluations and comments are confirmed. 
